# The Effects of Nitrogen and Phosphorus on Colony Growth and Zoospore Characteristics of Soil Chytridiomycota

**DOI:** 10.3390/jof8040341

**Published:** 2022-03-24

**Authors:** Deirdre G. Hanrahan-Tan, Linda Henderson, Michael A. Kertesz, Osu Lilje

**Affiliations:** 1School of Life and Environmental Sciences, The University of Sydney, Sydney, NSW 2006, Australia; michael.kertesz@sydney.edu.au; 2Department of Planning and Environment, Locked Bag 5022, Parramatta, NSW 2124, Australia; linda.henderson@environment.nsw.gov.au

**Keywords:** chytridiomycota, fungi, soil, nitrogen, phosphorus, biomass, zoospore, protein quantification, substrate attachment, motility

## Abstract

The Chytridiomycota phylum contributes to nutrient cycling and the flow of energy between trophic levels in terrestrial and aquatic ecosystems yet remains poorly described or absent from publications discussing fungal communities in these environments. This study contributes to the understanding of three species of soil chytrids in vitro—*Gaertneriomyces semiglobifer*, *Spizellomyces* sp. and *Rhizophlyctis rosea*—in the presence of elevated concentrations of nitrogen and phosphorus and with different sources of nitrogen. Colony growth was measured after 4 weeks as dry weight and total protein. To determine the impacts on zoospore reproduction, motility, lipid content, and attachment to organic substrates, 4- and 8-week incubation times were investigated. Whilst all isolates were able to assimilate ammonium as a sole source of nitrogen, nitrate was less preferred or even unsuitable as a nutrient source for *G. semiglobifer* and *R. rosea,* respectively. Increasing phosphate concentrations led to diverse responses between isolates. Zoospore production was also variable between isolates, and the parameters for zoospore motility appeared only to be influenced by the phosphate concentration for *Spizellomyces* sp. and *R. rosea*. Attachment rates increased for *G. semiglobifer* in the absence of an inorganic nitrogen source. These findings highlight variability between the adaptive responses utilised by chytrids to persist in a range of environments and provide new techniques to study soil chytrid biomass and zoospore motility by total protein quantification and fluorescent imaging respectively.

## 1. Introduction

The Chytridiomycota phylum of zoosporic true fungi (chytrids) is best known for the pathogenicity of certain species and their contributions to nutrient cycling in aquatic environments. Little is understood about how saprophytic chytrids adapt to fluctuating conditions in soil environments and what shifts occur in their biology when agricultural practices are applied. These practices include application to the soil of organic and inorganic forms of nitrogen and phosphorus fertilisers, which exposes chytrids to elevated nutrient levels both in the soil and in leachate entering rivers and streams. However, in addition to addressing the limitations of low phosphorus soil content and increasing agricultural yields, such fertilisation can also affect carbon cycling in the soil by accelerating or decreasing soil organic matter (SOM) turnover through the priming effect [1]. Long-term exposure and overuse of fertilisers can hence cause shifts in fungal population profiles [2]. Chytrids have been reported to respond to a variety of nitrogen or phosphorus sources [3,4], including at the genus level, however, the effects of increased nutrient supplies on different stages of the chytrid life cycle and growth have not been studied.

Chytrids are common in soil as both saprotrophs and parasites [5,6]. In Australia, numerous chytrid genera have been isolated from soils, including agricultural soils [7,8,9]. A consistent and substantial presence of chytrid phylotypes has been found in soils, and these taxa utilise diverse substrates [10,11] and contribute to nutrient cycling through the conversion of organic sources of nitrogen and phosphorus to inorganic forms such as nitrate (NO_3_^−^), ammonium (NH_4_^+^) and phosphates (HPO_4_^2−^, H_2_PO_4_^−^) for plant uptake. They also contribute to the degradation and mineralisation of cellulose, chitin, starch, protein and lipids from a range of plant and animal materials, thereby mobilising otherwise resistant organic matter [12,13,14]. Attachment, often including the penetration of host tissues by chytrid rhizoids, is likely to play a crucial role in substrate degradation [13,15,16,17].

Zoospores are a critical stage of the chytrid life cycle, enabling the colony to grow, disperse, and colonise new substrates. Zoospores in the order Rhizophlyctidales, including *Rhizophlyctis rosea*, contain lipid globules and other cellular organelles [18], all of which are assembled as the microbody-lipid globule complex [19]. In aquatic ecosystems, zoospore production and elemental content (defined as the amount of particulate organic carbon, nitrogen and phosphorus) have been shown to fluctuate according to N:P ratios [20]. The microbody-lipid globule complex acts as an endogenous energy store, ensuring the zoospore can move and seek a suitable substrate. It also establishes the zoospore as a highly nutritious food source, and zoospores are grazed on in aquatic and soil ecosystems by protists, particularly amoebae, ciliates, and invertebrates [21,22,23,24,25]. Zoospores, therefore, contribute to the transfer of nutrients, polyunsaturated fatty acid sources, cholesterol, phospholipids, glycogen, and nucleic acids to higher trophic levels [6].

The availability of the macronutrients nitrogen and phosphorus are of particular focus in the current study due to their importance in natural and agricultural soils. Whilst nitrogen for plants can be obtained either through bacterial nitrogen fixation or saprophytic activity, phosphorus is often limited in soils, especially in Australia due to low inputs from weathered rocks [26]. Fertiliser additions can reduce the abundance and richness of fungal communities [27] with the potential to increase the abundance of plant pathogenic fungi [2], whilst populations and the diversity of other plant beneficial fungi, such as arbuscular mycorrhizal fungi (AMF), may instead decrease [28,29]. A particular challenge for soil zoosporic fungi may be the spatial and temporal scarcity of utilisable nutrients as they compete with plants and other fungi for nutrients such as phosphorus and nitrate, particularly in agricultural soils [25].

Low microbial diversity, in addition to nutrient availability, can be an essential indicator of poor soil health. When an environment is exposed to stressors, abundant taxa dominate due to their ability to utilise a broader suite of nutrients and niches [30]. However, rare taxa, which often includes Chytridiomycota, are thought to provide a diverse taxonomic pool inevitably underpinning the microbial community’s resilience and resistance as a whole [30,31]. Therefore, rather than disregarding the roles of soil chytrids due to their low abundance in various soils [32], it is important we continue to determine their basic biological functions to continually improve our understanding and determine their genomic capabilities [33].

The objective of this study was to determine changes in biomass (dry weight and total protein), reproduction (zoospore quantification), zoospore lipid quantification and motility (velocity, meandering index, and displacement), and substrate attachment, of three soil chytrids following exposure to different concentrations of nitrogen and phosphorus and different sources of nitrogen. We hypothesised that when nutrient concentrations increase, chytrid biomass and zoospore production would increase, but further increases in nutrient concentrations may have a negative influence on growth. Furthermore, these parameters would also depend on the ability of the fungi to utilise various nitrogen sources. Here, we report that *Spizellomyces* sp. can adapt and utilise different nitrogen sources and high concentrations, but *R. rosea* is restricted in its utilisation of nitrate, and higher concentrations can impede colony yields for both *R. rosea* and *G. semiglobifer*. Phosphorus was expected to be the limiting nutrient for zoospore lipid and motility outcomes. This postulate was observed for *Spizellomyces* sp. and *R. rosea* cultures but not for *G. semiglober* in the current study. These observations are likely to reflect an ecological strategy—stress (S-adaptive), combative (C-adaptive), ruderal (R-adaptive) [34]—adopted by the chytrids at various stages of their life cycle. The ability of soil chytrids to exploit different sources and concentrations of nitrogen and phosphorus, surviving both scarcity and elevated concentrations, may help predict their ability to thrive in soil environments.

## 2. Materials and Methods

### 2.1. Selection and Maintenance of Isolates

Three species of zoosporic true fungi from the University of Sydney Zoosporic fungi culture collection were chosen for this study. These isolates all originate from soils in New South Wales (NSW) and were selected for this study because they display similar growth cycles and highlight both intra- and inter-genus differences in responses to nutrient treatments. The isolates represent two orders within the Phylum Chytridiomycota. *Rhizophlyctis rosea* (AUS 13: Order Rhizophlyctidales) from Sydney, *Gaertneriomyces semiglobifer* (Mar CC2: Order Spizellomycetales) and *Spizellomyces* sp. (Dec CC 4-10Z: Order Spizellomycetales) isolated from cropping soils in Narrabri, north-western NSW (−30°16′47.4″, 149°47′43.2″) (Table 1).

### 2.2. Treatment Media for Experimental Testing

Chytrid Synthetic Medium (CSM) is routinely used for in vitro experimental work with chytrids [3] and was used in the current study to prepare the treatment media with minor modifications including the use of Na_2_MoO_4_·2H_2_O instead of (NH_4_)_6_Mo_7_O_24_. Thiamine (0.5 mM) is reported as an optional addition [3] but was excluded in this experiment as it was determined to be non-essential for the growth of the selected isolates. Chytrid cultures were incubated on solid starvation CSM (Table 2) (24 °C for a minimum of one week) to ensure cultures were depleted of internal nitrogen and phosphate stores prior to inoculation on the treatment media. 

The pH of each CSM liquid treatment medium was measured using a Mettler Toledo EL2 pH probe and averaged a pH of 7.67 ± 0.55 SD. Notably, the 0 mM nitrogen treatment had a pH of 8.02 and the pH of the 0 mM phosphate treatment was 5.45.

### 2.3. Soil Extract Medium

Soil was collected from the gardens surrounding Cadigal Green at the University of Sydney, Darlington NSW 2008, Australia (−33°53′23.8″, 151°11′31.4″). The soil sample collected contained 0.10 % total Kjeldahl nitrogen and 31 mg/kg of Colwell phosphorus as determined by Nutrient Advantage, Werribee VIC 3030, Australia (https://www.nutrientadvantage.com.au/, accessed on 18 May 2019). Soil extracts for use in growth media were prepared according to the German Collection of Microorganisms and Cell Cultures protocol [35]. Soil (100 g) was dried overnight at 40 °C to constant weight. In 1 L of DI H_2_O, 400 g of air-dried soil was sterilised (121 °C, 60 min), then allowed to sediment and cool to room temperature. The supernatant was centrifuged (2500 rpm, 20 min) using a Clements 2000 centrifuge (Clements). The supernatant was collected. For solid medium, 1.5% *w*/*v* of agar was added. For liquid media used in the dry weight biomass experiments, 22 mM glucose was added to ensure naturally low soil carbon concentrations were not a limiting factor. The pH was adjusted to 6.8–7.0 and then sterilised under the same conditions. 

### 2.4. Dry Weight Biomass

Chytrid cultures were prepared on plates by incubating on solid starvation CSM (Table 2) (24 °C for a minimum of one week). Each Petri dish was then flooded with 3 mL of DI H_2_O for 2 h to promote sporulation. The resulting spore suspension was used to inoculate liquid medium (21 mL) in 50 mL Falcon tubes with 10^3^ zoospores mL^−1^. Samples then underwent static incubation in the dark (24 °C, 4 weeks).

After 4 weeks, chytrid cultures grown in liquid media were resuspended to form a homogeneous solution by physically detaching the cultures that had adhered to the Falcon tube walls with a sterile cell scraper. An aliquot (1 mL) of each replicate was transferred to a sterile 2 mL Eppendorf for protein analysis. As described by Henderson (2018) [17], the remaining solution was centrifuged in a Clements 2000 centrifuge (Clements, Rydalmere NSW 2116, Australia) (2500 rpm, 30 min). The supernatant was discarded, and the remaining pellet was resuspended in 20 mL of DI H_2_O before re-centrifuging (2500 rpm, 15 min). The majority of the DI H_2_O was removed, and the remaining pellet was rinsed into pre-weighed foil cups for drying (65 °C, 24 h). The dried dishes were weighed on a Mettler Toledo TLE303 balance (Mettler-Toledo Ltd., Port Melbourne VIC 3207, Australia).

### 2.5. Protein Quantification

Total protein quantification was based on a modified Lowry assay [36]. Total proteins in the culture (extracellular and intracellular) were precipitated by the addition of 0.2 mL 3 M trichloroacetic acid to each 1 mL sample of liquid culture and frozen at −20 °C. Samples were thawed and the cells and precipitated protein were collected by centrifugation (Centrifuge 5410, Eppendorf South Pacific Pty Ltd., Macquarie Park NSW 2113, Australia) (14,000 rpm, 60 min). The supernatant was discarded, and the pellets were resuspended in 0.6 M NaOH (0.6 mL) and incubated overnight at room temperature on a shaker. Undissolved cell debris was removed by centrifugation (14,000 rpm, 60 min) and the supernatant was transferred to clean 1.5 mL microcentrifuge tubes. To the redissolved samples, 0.6 mL Lowry Reagent D (Sodium carbonate 13% *w*/*v*, Sodium potassium tartrate 4% *w*/*v*, Copper (II) sulphate pentahydrate 2% *w*/*v*) in the ratio 100:1:1 was added and incubated for 10 min, at which point 0.2 mL Folin-Ciocalteu reagent (Merck Life Science Pty Ltd, Bayswater VIC 3153, Australia) (diluted 1:5) was added. A negative control for soil extract was also analysed to account for residual protein in the medium. Bovine serum albumin (0–200 μg) was used as a standard. Absorbance was measured at 625 nm on a SPECTROstar Nano microplate reader (BMG Labtech Pty Ltd., Mornington VIC 3931, Australia).

### 2.6. Chytrid Zoospore Quantification

Sporulation on solid agar treatments (Table 2) was induced as per Henderson [37], except that 5 mL of DI H_2_O was used. The zoospores were enumerated using a haemocytometer at 4- and 8-weeks.

### 2.7. Microscopic Analysis of Zoospore Motility and Lipid Quantification

Protocols for fluorescent staining of the lipid content of zoospores using the lipophilic Nile Red stain [23,37] were adapted for the current study.

Culture plates were flooded for 1 h with 5 mL DI H_2_O, and 999 μL of the resulting zoospore suspension was collected into an Eppendorf containing 1 μL of the Nile Red solution (4 mg mL^−1^ in dimethylsulfoxide). The suspension was then incubated in the dark at room temperature for approximately 10 min.

Stained preparations of zoospore suspensions were imaged on a glass slide using the Nikon A1R Advanced Confocal Microscope with either the Plan Apo VC 20× DIC N2 or Apo LWD 40× WI λS DIC N2 objective. The photomultiplier tube gain and offset were kept constant across all experiments. Time-lapse sequences were captured at either 1 or 10 millisecond intervals for a duration of 1 min, or at 1 s intervals over 2 min (laser excitation 560 nm and fluorescence emission 450 nm).

Time-lapse data were analysed with Volocity 6.3 (PerkinElmer, Melbourne VIC 3150, Australia). The fluorescence channel, using Red-Green-Blue intensity, was used to identify zoospores and then segment each one within each time series experiment. The movement of the zoospores was analysed using the “Track Object” algorithm. Gaps between tracks of the same zoospore were closed where possible. Each identified track and object were analysed separately and the averages of these were determined for each time-lapse. Additionally, within each frame, the fluorescence intensity of individual zoospores was analysed and recorded.

### 2.8. Attachment to Substrates

Each attachment plate consisted of a 70 mm diameter circular rubber (Kadink foam sheets) disc in which five 6 mm diameter holes had been punched in. Rubber discs were sterilised according to Henderson et al. 2018 [17], except overnight drying was at 80 °C. Rubber disks and Petri dishes were then UV sterilised for 20 min. Three mm diameter circles of snake skin (keratin) and lens paper (cellulose) were prepared using a hole punch and autoclaved (121 °C, 20 min). Prepared Petri dishes containing the rubber discs were filled with 15 mL of each of the treatment liquid mediums. Inoculation volumes were adjusted to ensure 400 zoospores were added per hole. In the instance that zoospore concentrations were too low, double or five times this inoculation volume was used. A solid PYG plate was similarly inoculated with the same concentration of zoospores for each replicate to determine if zoospores were successfully sampled. The prepared baits—snake skin for *Spizellomyces* sp. and *G. semiglobifer*, or lens paper for *R. rosea* [17]—were added to each inoculated hole then incubated (5 days, 24 °C). The number of large developed chytrids attached to each bait was counted by visualising the 3 mm circumference of the bait with an Olympus CK 2 inverted microscope using an LWD C A20PL 20× objective. Within the field of view, the area of the liquid medium surrounding the bait was also surveyed and the number of unattached chytrids was recorded to determine the percentage attachment. 

### 2.9. Statistical Analysis

Statistical analyses were performed using two-way analysis of variance (ANOVA) and Tukey’s correction for multiple comparisons using GraphPad Prism Version 8.0.2 for Windows (GraphPad Software, San Diego, California USA). Comparisons resulting in a *p* value < 0.05 were considered significant. Nitrogen and phosphorus concentration series were analysed separately and CSM and soil extract media were compared against both the nitrogen and phosphorus treatments. Depending on the experiment, the analyses were compared against the factor of incubation time (4 and 8 weeks) and/or between species (*G. semiglobifer, Spizellomyces* sp. and *R. rosea*).

## 3. Results

### 3.1. Chytrid Biomass Measured as Dry Weight (mg)

#### 3.1.1. Cell Yields Varied between Chytrid Species When Exposed to Different Sources and Concentrations of Nitrogen

Ammonium as a sole nitrogen source was utilised at various concentrations by all three chytrids (Figure 1a,b). *Rhizophlyctis rosea* cultures significantly increased in dry weight when grown on CSM or 1.5–20 mM ammonium treatment media compared to organic nitrogen (L-Ala/L-Met) and the nitrate treatments (Figure 1c). However, 20 and 50 mM concentrations of ammonium resulted in significantly lower dry weight biomass compared to CSM. This decrease was also observed in *G. semiglobifer* dry weights (Figure 1a). *Rhizophlyctis rosea* growth with all tested concentrations of nitrate was similar to growth in the absence of nitrogen (0 mM nitrogen) suggesting that *R. rosea* is less capable of assimilating nitrate. Growth on organic nitrogen (L-Ala/L-Met) and soil extract medium was comparable to growth in the absence of nitrogen (0 mM nitrogen) for both *R. rosea* and *Spizellomyes* sp. cultures (Figure 1b,c), whereas *G. semiglobifer* cultures grown on CSM, soil extract or organic nitrogen sources had similar dry weight biomass (Figure 1a). *Spizellomyces* sp., unlike *R. rosea* and *G. semiglobifer*, utilised all concentrations of nitrogen (1.5–50 mM) despite being limited to a single inorganic nitrogen source (ammonium or nitrate) (Figure 1b).

#### 3.1.2. All Chytrids Increased in Dry Weight When Grown in Various Concentrations of Phosphate

*Spizellomyces* sp. cultures grown on treatment media with higher concentrations of phosphate (5 and 20 mM) produced increased dry weight biomass compared to the absence of phosphate or soil extract medium (Figure 1b). No significant changes in dry eight were observed across the various phosphate concentrations for *G. semiglobifer* colonies Figure 1a). A phosphate concentration of 5 mM resulted in the highest dry weight biomass for *R. rosea* cultures (F_5, 75_ = 41.97, *p* < 0.05) (Figure 1c).

### 3.2. Total Protein (µg mL^−1^ BSA Equivalents) Is a Reliable Method for the Measurement of Chytrid Biomass

Total protein in each chytrid culture was measured in addition to dry weight to determine whether quantitative biochemical assays, as a measure of fungal biomass, are suitable for working with chytrids. Overall, similar trends were observed between the two measures of biomass. 

#### 3.2.1. Chytrids Showed Diverse Abilities to Grow in the Presence of Different Nitrogen Treatments as Measured by Total Protein

Changes in the total protein yields for *G. semiglobifer* cultures (Figure 2a) grown in different treatment media were more pronounced compared to dry weight measurements (Figure 1a). However, 5 mM ammonium treatment media produced the highest biomass content across both methods. The growth of *Spizellomyces* sp. in CSM and all concentrations of nitrogen (1.5–50 mM nitrate/ammonium) was similar (Figure 2b), reflecting the observed dry weight trend (Figure 1b).

*Rhizophlyctis rosea* total protein yields (Figure 2c) also reflected the dry weight biomass results (Figure 1c). Growth in the nitrate treatments or in the absence of nitrogen (0 mM nitrogen), was significantly lower compared to yields when cultures were incubated in ammonium treatments (Figure 2c). Yields were also reduced after growth in the soil extract medium and organic nitrogen (L-Ala/L-Met) treatment. Additionally, *R. rosea* total protein yields decreased when exposed to higher concentrations of ammonium (5, 20 and 50 mM) compared to growth in the 1.5 mM ammonium treatment.

#### 3.2.2. The Maximum Yield of Total Protein Varied between Chytrids Depending on Phosphorus Concentrations

Total protein yields from *R. rosea* cultures were significantly higher when grown on 5–20 mM phosphate treatments compared to growth on soil extract and 0 mM phosphate (Figure 2c). A decrease in the yield of *R. rosea* total protein was observed as the phosphate concentration increased. *Spizellomyces* sp. cultures instead increased in total protein with increasing concentrations of phosphate where yields from growth in 20 mM phosphate treatments were higher compared to growth on CSM, 5 and 10 mM phosphate treatments (Figure 2b). Total protein yields of *G. semiglobifer* did not significantly change with exposure to different phosphate concentrations (Figure 2a) as observed in the dry weight measurements (Figure 1a).

### 3.3. Zoospore Production Increased under Nutrient Deprivation for Some Chytrids Whilst Others Benefited from Increased Nutrient Availability

Higher zoospore counts of *Spizellomyces* sp. were recorded after growth in CSM compared to cultures grown on soil extract, organic nitrogen (L-Ala/L-Met) and the nitrogen series (0–50 mM ammonium nitrate) at both 4 and 8 weeks (Figure 3). Growth on the various phosphate treatments produced similar numbers of *Spizellomyces* sp. zoospores.

*Gaertneriomyces semiglobifer* cultures produced significantly higher zoospore yields in the absence of a nitrogen source (0 mM nitrogen) at both 4 and 8 weeks of incubation compared to growth on CSM, soil extract, and all other nitrogen treatments. Growth on CSM and soil extract medium produced significantly more zoospores after 4 weeks incubation than when *G. semiglobifer* cultures were grown in the presence of higher (5, 10 and 20 mM) phosphate concentrations.

Cultures of *R. rosea* grown on organic nitrogen (L-Ala/L-Met), soil extract, or 0 mM nitrogen treatment medium produced significantly less zoospores than when grown on CSM. Zoospore yield significantly increased when *R. rosea* was exposed to elevated phosphate concentrations (10 and 20 mM).

### 3.4. Zoospore Motility and Lipid Production Differed between Chytrids Depending on Nutrient Source and Concentration

No significant changes in lipid content (mean fluorescence) and motility (velocity, displacement and meandering index) were observed when *G. semiglobifer* and *R. rosea* zoospore were cultured in different nitrogen treatment media (Figure 4). The lipid content of *Spizellomyces* sp. when incubated on CSM and organic nitrogen (L-Ala, L-Met) treatments was significantly lower than the lipid content after incubation on 5 mM ammonium nitrate.

Zoospores of both *R. rosea* and *Spizellomyces* sp. grown in the absence of phosphate had significantly higher lipid content than those from sporangia grown in all concentrations of phosphate and also CSM for *Spizellomyces* sp. Although measurements of zoospore motility resulted in minimal significant findings, they further indicated that despite higher lipid contents when incubation in the absence of phosphate, the zoospores of *Spizellomyces* sp. and *R. rosea* travelled more slowly within a confined area and with minimal displacement, compared to zoospores from cultures grown in CSM or 5 and 20 mM phosphate treatments. Zoospores of *G. semiglobifer* sporangia did not differ in lipid content, velocity, displacement or meandering index with changes in the concentration of phosphate.

The software, Volocity 6.3 (PerkinElmer, Melbourne VIC 3150, Australia), successfully tracked individual chytrid zoospores to visualise and measure their motility (Figure 5). 

### 3.5. Attachment to an Organic Substrate Changed Significantly in Response to Different Nitrogen Sources

The percentage attachment of *G. semiglobifer* to snake skin increased significantly when exposed to soil extract, organic nitrogen (L-Ala/L-Met) and 0 mM nitrogen compared to growth in 5 and 50 mM ammonium nitrate or CSM (Figure 6). Attachment of *Spizellomyces* sp. to keratin (snake skin) was not significantly altered depending on the available nitrogen source (Figure 6). No significant differences in the percentage of attachment were observed for *R. rosea* cultures grown in different treatments. In comparison to *Spizellomyces* sp., *R. rosea* had a significantly higher percentage attachment at 0 mM phosphate and organic nitrogen (L-Ala/L-Met) (Figure 6).

## 4. Discussion

Differentiation of nitrogen and phosphorus resource utilisation occurs at different stages in the life cycles of the three chytrids isolated from soil environments of NSW. This study demonstrates that elevated nutrient supplies and the source of nitrogen impact the biology of soil chytrids in various ways. Whilst each of the studied chytrids responds uniquely, some traits are shared by both species in the Spizellomycetales Order (Table 1), including the utilisation of nitrate (Figure 1 and Figure 2). Zoospore production and lipid quantification studies indicated that *R. rosea* and *Spizellomyces* sp. responded with similar patterns compared to the two Spizellomycetales (*Spizellomyces* sp. and *G. semiglobifer*) (Figure 3 and Figure 4). Alternative chytrid experimental methods are also presented, including the use of the fluorescent dye Nile red to stain chytrid lipids and track zoospore motility (Figure 4) and a technique for measuring chytrid biomass by total protein quantification. Total protein quantification to determine microbial biomass is more robust and reliable compared to traditional dry weight measurements, which lose sensitivity with smaller yields of mass [38]. Further development of these techniques will benefit future chytrid investigations as many of the methods used for common fungi are not applicable for chytrids. For example, ergosterol quantification is used to determine fungal colony biomass of fungal colony, however, the sterol is absent from chytrid cell membranes [10,39,40]. The *in vitro* observations from this study provide a preliminary insight into the direct effects of elevated nitrogen and phosphorus concentrations or selected nitrogen sources on the colony and zoospores of saprotrophic soil chytrids.

At different stages of their life cycles, each of the chytrid isolates investigated adopt an ecological strategy—ruderal (R-selected) or stress-tolerant (S-selected) [34]. This is known for other fungi as well—AMF, for example, uses high nutrient availability to maximise reproductive potential, increasing spore density as nitrogen and phosphorus increase [41]. Aquatic chytrids are known to frequently exhibit ruderal strategies, quickly adapting to nutrient inputs that result in chytrid epidemics [21,22]. The population density of *R. rosea* has previously been observed to increase rapidly in soil environments when favourable conditions are present [42]. In the current study, *R. rosea* zoospore production increased with increasing phosphate concentrations (Figure 3) therefore showcasing ruderal (R-selected) adaptive strategies to exploit temporally and spatially variable phosphate supplies [25]. Greater dependence on adaptive, stress-tolerant (S-selected) strategies are evident when nitrogen resources are scarce or less able to be utilised. *Rhizophlyctis rosea* colony sizes were smaller overall (Figure 1 and Figure 2), and sporulation appeared either delayed or impeded (Figure 3). This response may redirect available resources to prioritise phenotypic modifications of thallus structures and colony pigmentation, thereby prioritising resistant structures. On the other hand, *G. semiglobifer* cultures produced significantly more zoospores when grown in the absence of a nitrogen source, analogous to e.g., *Geotrichum candidum* and *Mucor racemosus*, which also increase arthrospore numbers under nitrogen deprivation in vitro [43]. As hypothesised, this may be an alternative stress adaptation (S-selected) utilised by some species to prioritise colony dispersal when resources are scarce. This is further supported by the higher rate of attachment of *G. semiglobifer* to keratin when an inorganic source of nitrogen—particularly ammonium—is either absent or low, such as in the soil extract, organic nitrogen (L-Ala/L-Met) and 0 mM nitrogen treatments (Figure 6).

The biomass of *Spizellomyces* sp. grown in the presence of nitrate was unexpectedly equal to that of growth in ammonium (Figure 1 and Figure 2) and the total protein increased as phosphate concentrations increased (Figure 2) suggesting ruderal, copiotrophic strategies. Previous investigations indicated that after one week of incubation, ammonium would be preferred and growth in nitrate would yield reduced biomass [3]. This was the case for the biomass of *G. semiglobifer,* where growth in the presence of nitrate as the sole nitrogen source was consistently lower at all concentrations and across the two biomass measurements (Figure 1 and Figure 2). These findings of soil chytrids in the current study highlight not only differences in adaptations among the three species, but also shifts in each chytrid’s strategic responses [34] when the availability of resources fluctuates.

This study is not able to provide insight on whether combative (C-selected) strategies are also utilised, but it is likely that in vivo soil environments would drive chytrids to adopt combative mechanisms at various stages of their life cycles. The higher zoospore production rates of *Spizellomyces* sp. and dry weight biomass of *R. rosea* when incubated in CSM indicates that, for these chytrids, both organic and inorganic nitrogen sources may be required to support optimal reproduction and colony growth. However, the low growth of the cultures in soil extract when compared to CSM (Figure 1) emphasises the favourable conditions provided in vitro where the constant changes in moisture and nutrient availability within soil environments is absent [44], as well as competition, predation, and other stressors including desiccation. 

*Spizellomyces* sp. may contain a broader suite of metabolic pathways aiding the transformation of commonly inaccessible nutrients, thereby building resilience to a range of environments. Within Dikarya, nitrogen regulatory pathways are relatively well characterised [45] and the nitrogen metabolite repression mechanism regulates the use of nitrate as a resource for fungi if other sources such as ammonium and glutamate are absent [46,47]. The presence of these preferred sources (ammonium and glutamate) represses the synthesis of nitrate reductase in these fungi [46]. Our dilemma is that models for fungal nitrogen pathways have been limited to yeasts and filamentous fungi [45,48,49] and the gene homologs which form the nitrate assimilation cluster have not been found in Chytridiomycota [50,51].

Monocentric chytrids, including those tested in this study, are understood to be incapable of accessing phosphorus from DNA as an organic source, suggesting their inability to produce phosphomono- and phosphodiesterases, which are present in filamentous fungi [4]. However, phytate, as an organic phosphate source, may be accessible by some monocentric chytrids including *R. rosea* [4]. It is thought that chytrid populations are limited in uncultivated soils where inorganic phosphate salts are scarce [52]. Whilst there are currently no available studies to confirm this, it could be expected that as phosphate fertiliser applications increase, so too will some monocentric chytrid species. Although the abundance of Chytridiomycota in soils is often low, more work is necessary if we want to understand the intricacies and diversity of chytrid nitrogen and phosphorus metabolism and hence determine their potential contributions to soil microbial ecology [30,31,33]. 

The growth of *R. rosea* colonies was severely reduced in the presence of nitrate. Measurements of total protein and dry weight suggest that *R. rosea* is unable to assimilate nitrate (Figure 1 and Figure 2). It would be interesting to observe whether growth in nitrate as a sole nitrogen source induces a similar response as growth in the absence of nitrogen with regard to zoospore production (Figure 3). Secondary metabolites and alternative stress responses may be induced by some chytrids during extreme stress. *Rhizophlyctis rosea, Spizellomyces puntatas* and *Chytridiales* spp. are known to contain rhodopsins and carotenoids [53,54], which can be observed as orange pigmentation in *R. rosea* cultures. Recent studies have suggested that retinal chromophores may contribute to the nutrition of some fungi by enabling adaptations under stressful conditions including low nutrient availability [55,56].

High genetic diversity, predicted within certain enzyme groups, suggests *R. rosea* has acquired and retained a broad collection of carbohydrate-active enzymes [33]. The versatility and adaptability of *R. rosea* to utilise different substrates may be demonstrated by its large, diverse secretome, containing cellulolytic, xylanolytic, pectinolytic, mannan and starch degrading enzymes [33]. The increase in attachment to keratin by *G. semiglobifer*, when nitrogen is low, foreshadows the ability of this chytrid to dismantle the secondary protein structure of keratin and mineralise the nitrogen present, such as from cysteine residues, into utilisable sources. There is still much to learn about these mechanisms and their contribution to fungal nutrition, particularly in disturbed environments.

As expected, phosphorus is a limiting nutrient determining the outcome of chytrid zoospore production and motility. Spore size is not a common measurement or focus in studies of microorganisms [57], but some reports have begun to discern its relevance for some fungi. Observations of AMF, for example, have found a trade-off between spore output and spore size, where species producing large-sized spores had lower spore production rates when compared to small-spored species [57]. Similarly, the size (defined as the nitrogen and phosphorus content) and quantity of zoospores of the aquatic parasite *Rhizophydium megarrhizum* also shifted in response to nitrogen and phosphorus ratios such that low N:P treatments led to larger spores, but these were produced in a smaller quantity [20]. In the current study, when *Spizellomyces* sp. and *R. rosea* cultures were grown in the absence of phosphate, resources appear to be directed towards zoospore lipid production measured as fluorescence (Figure 4). Furthermore, the lipid content of *R. rosea* zoospores was negatively correlated to zoospore production of these cultures (Figure 3 and Figure 4). Although zoospore lipid content cannot be correlated to spore size [58], the changes in lipid fluorescence and therefore lipid content observed in the current study in comparison to zoospore formation suggest a similar trade-off. Lipids present in the zoospores of other organisms such as the giant kelp *Macrocystis pyrifera* [58] and the alga *Saccharina latissimi* [59] are known to provide the required energy for motility and are therefore depleted over the zoospores’ lifespan. Certain species, including *R. rosea*, may produce fewer but more nutrient-rich zoospores to increase the chance of survival and extend the motile phase of their zoospores, thereby enabling the colony to seek substrates further afield in adverse conditions.

Motility measurements for *R. rosea* and *Spizellomyces* sp. indicated that although zoospores had higher lipid contents when incubated in the absence of phosphate, they travelled slowly, within a confined area, and with minimal displacement (Figure 4). This contradicts our original expectation that a higher lipid content in zoospore microbody-lipid globule complexes would allow for greater zoospore activity. *Spizellomyces* sp. and *R. rosea* also had increased lipid contents following incubation with low levels of ammonium nitrate. Exposure to different phosphate concentrations (5, 10, and 20 mM phosphate) resulted in the opposite response in *G. semiglobifer,* where significant reductions in zoospore production were found after 4 weeks of incubation when compared to growth on CSM (3.44 mM phosphate) and soil extract (31 mg/kg of Colwell phosphorus) (Figure 3). Instead, a significant increase in produced zoospores occurred when cultures were grown in the absence of nitrogen (Figure 2). Without the basic provision of nutrients essential for cellular maintenance, some species may prioritise dispersal by sporulation as a means of survival [60].

Investigations of zoospore lipids and production have predominantly focused on aquatic and parasitic chytrids due to the contributions and impacts of these chytrids on food-web dynamics and host infectivity [24,61,62]. Attention to the prevalence and responses of soil chytrids is equally important as they contribute to decomposition but may also provide a crucial source of food for grazers and predators in the soil food web. These roles in soil are yet to be clarified. This study, with refinement, presents a potential route for investigating the parameters of zoospore motility (lipid content, velocity, displacement and meandering index) via Nile red staining and analysis with programs like Volocity 6.3 (PerkinElmer, Melbourne VIC 3150, Australia) to better understand chytrid zoospore biology (Figure 5).

It is important that we continue to observe saprotrophic soil chytrids, however discreet their presence may be. Recognition of the roles and diverse nature of early diverging lineages of fungi including the Chytridiomycota is widening, and researchers are beginning to elucidate the pivotal roles of chytrids in nutrient cycling and trophic upgrading [25,33,63]. This study contributes to our primary understanding of soil chytrid nutrition under increased phosphorus and nitrogen availability and in the presence of different nitrogen sources. It is clear from our findings that variable responses from the chytrids investigated reflect diverse metabolic regimes that enable survival in adverse environments. Genetic sequence data for chytrids is still lacking, thus limiting our understanding of fungal population dynamics within various environments [10,64]. Strengthening molecular methods to investigate gene and protein regulatory systems in chytrids is pertinent to discovering unique enzymes and secondary metabolites with the potential for biotechnological applications [33]. Such applications could include bioprospecting for microbial keratinases to remove and recycle excess organic waste [65,66,67] or harnessing the ability of chytrids to degrade agricultural lignocellulose waste to enhance biofuel yields and prevent anoxic waterways [14,68]. Chytrids are omnipresent and their persistence in harsher environments highlights relatively unexplored genetic adaptations, contributing to the ecology and health of diverse environments.

## Figures and Tables

**Figure 1 jof-08-00341-f001:**
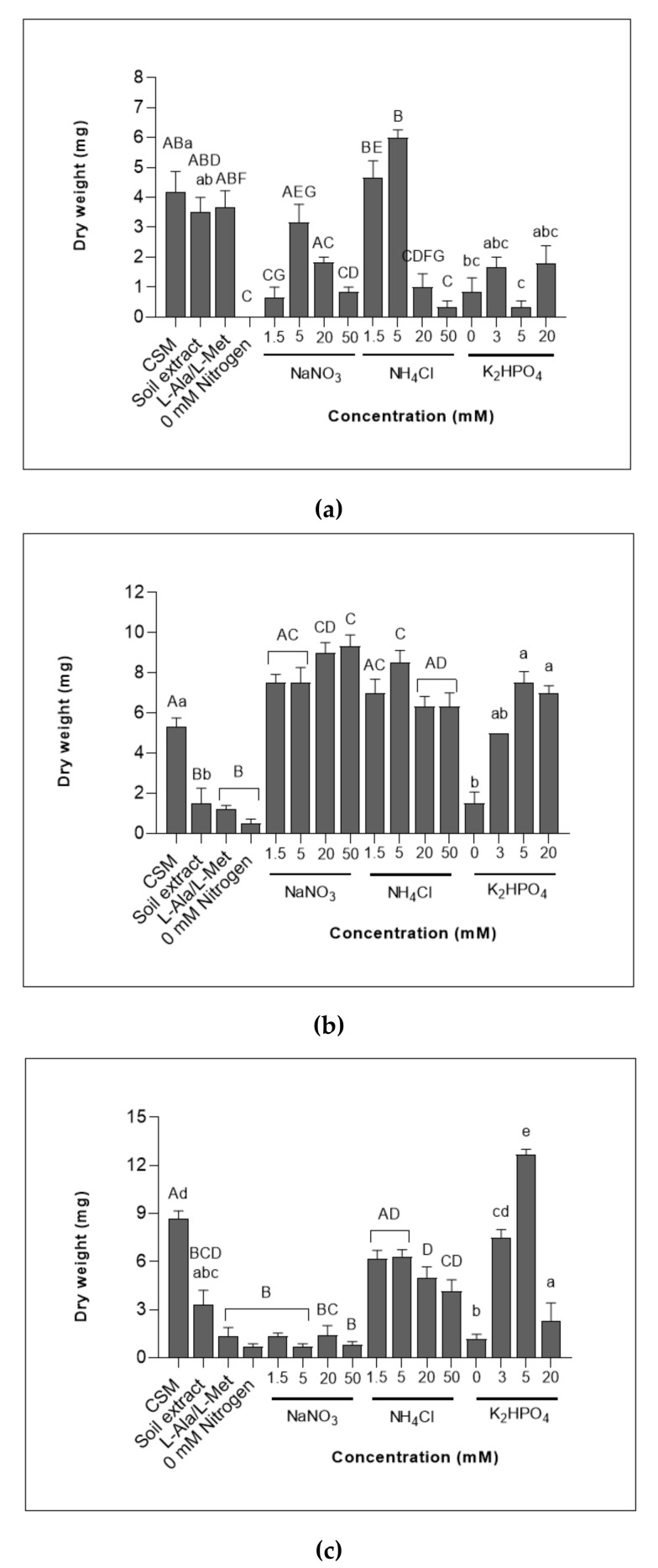
Biomass measured as dry weight (mg) at 4 weeks of growth. (**a**) *G. semiglobifer* (**b**) *Spizellomyces* sp. (**c**) *R. rosea*. Error bars are Standard Error of the Mean (SEM). Significant differences in the nitrogen series are denoted by capital-letters (A–G) (F_11, 175_ = 39.75, *p* < 0.05) and lower-case letters (a–e) are used for findings in the phosphorus series (F_5, 75_ = 41.97, *p* < 0.05).

**Figure 2 jof-08-00341-f002:**
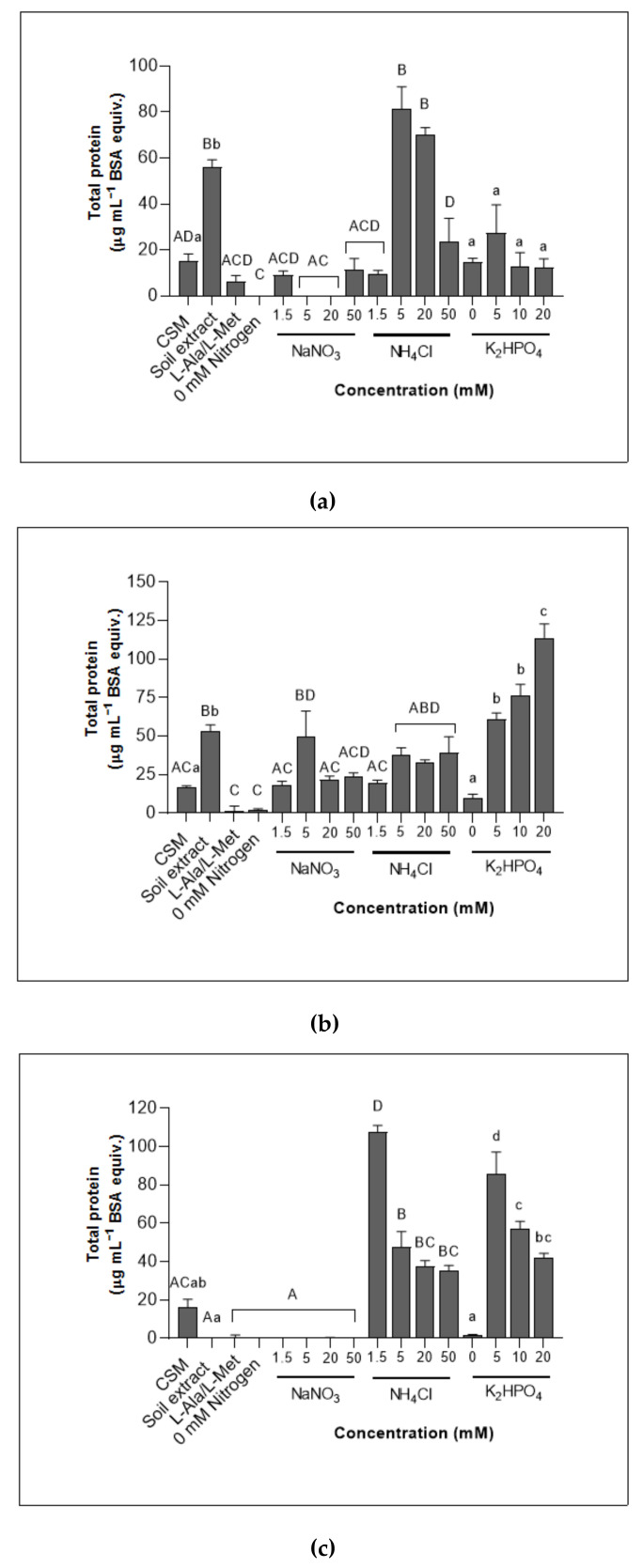
Biomass measured as total protein (μg mL^−1^ BSA equivalents) at 4 weeks of growth. (**a**) *G. semiglobifer* (**b**) *Spizellomyces* sp. (**c**) *R. rosea*. Error bars are SEM. Significant differences in the nitrogen series are denoted by capital letters (A–D) (F_11, 177_ = 48.69, *p* < 0.05) and lower-case letters (a–d) are used for findings in the phosphorus series (F_5, 90_ = 42.52, *p* < 0.05).

**Figure 3 jof-08-00341-f003:**
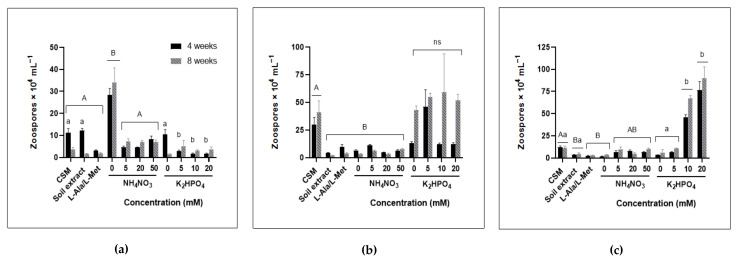
Zoospore haemocytometer quantification at 4 and 8 weeks. (**a**) *G. semiglobifer.* Significant differences were found between treatments in the nitrogen series (F_6, 69_ = 39.32, *p* < 0.05) and phosphate series (F_5, 59_ = 5.465, *p* < 0.05) (**b**) *Spizellomyces* sp. Significance was found in the nitrogen series (F_6, 68_ = 22.11, *p* < 0.05), but not the phosphate series (ns) (F_5, 56_ = 3.371, *p* > 0.05) (**c**) *R. rosea*. Significance was found in both the nitrogen (F_6, 70_ = 15.94, *p* < 0.05) and phosphate series (F_5, 60_ = 96.17, *p* < 0.05). Error bars are SEM. Significant difference in the nitrogen series are identified with capital letters (A, B) and lower-case letters (a, b) indicate significance between the phosphate treatments.

**Figure 4 jof-08-00341-f004:**
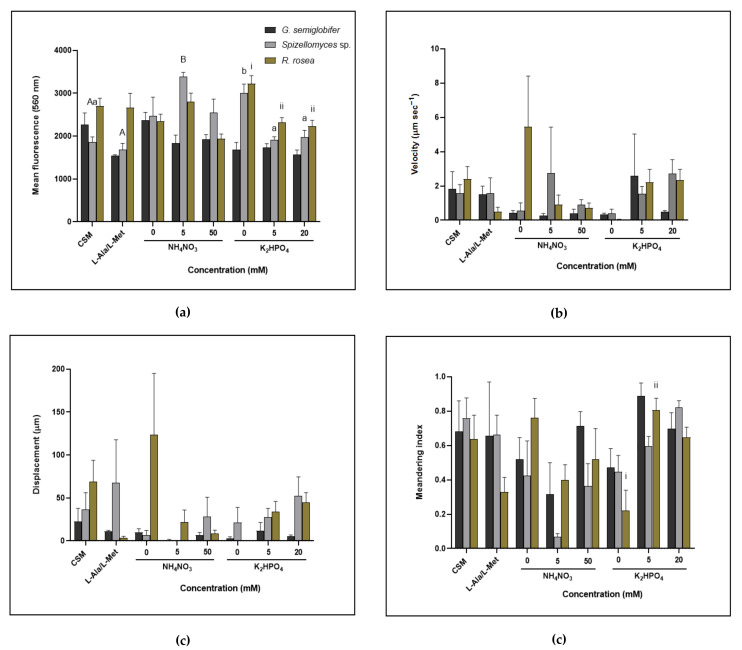
Microscopic analysis of chytrid zoospores at 8 weeks. (**a**) Mean fluorescence of lipid-stained zoospores. Significant differences were found when nitrogen treatments were compared (F_4, 57_ = 4.047, *p* < 0.05) and the phosphate treatments (F_3, 51_ = 12.52, *p* < 0.05) (**b**) Zoospore velocity (μm s^−1^). No significant differences were observed in the nitrogen (F_4, 56_ = 0.719, *p* > 0.05) or phosphate series (F_3, 50_ = 2.841, *p* > 0.05) (**c**) Displacement (μm) of zoospores. No significant differences were observed in the nitrogen (F_4, 56_ = 1.168, *p* > 0.05) or phosphate series (F_3, 51_ = 2.535, *p* > 0.05) (**d**) Meandering index of zoospores determined as the displacement rate divided by the velocity (μm s^−1^). A meandering index of 1.0 indicates the object travelled on a linear trajectory. No significant differences were observed in the nitrogen series (F_4, 56_ = 4.201, *p* > 0.05) but significance was observed in the phosphate series (F_3, 51_ = 9.504, *p* < 0.05). Error bars are SEM. Alphabetical letters distinguish differences found within *Spizellomyces* sp. cultures (A, B = nitrogen series; a, b = phosphate series), and roman numerals represent significance found within *R. rosea* cultures (i, ii = phosphate series).

**Figure 5 jof-08-00341-f005:**
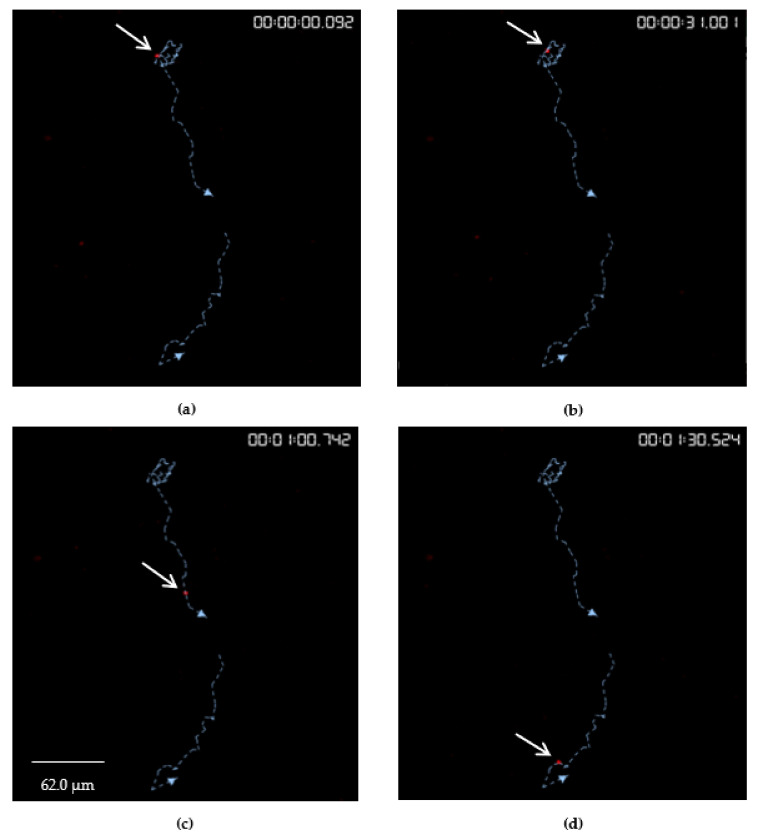
Nile red-stained zoospores of *Spizellomyces* sp. collected from cultures grown on 5 mM phosphate at 8 weeks. Four image sequences over the course of 1.5 min with 1 s intervals (**a**–**d**). Two tracks (blue dotted lines with arrow-heads) were identified for a single zoospore (white arrows). The time-lapse sequence had a mean fluorescence of 2041.5 at 590 nm, mean velocity of 2.4 μm s^−1^, mean displacement of 33.75 μm and mean meandering index of 0.74.

**Figure 6 jof-08-00341-f006:**
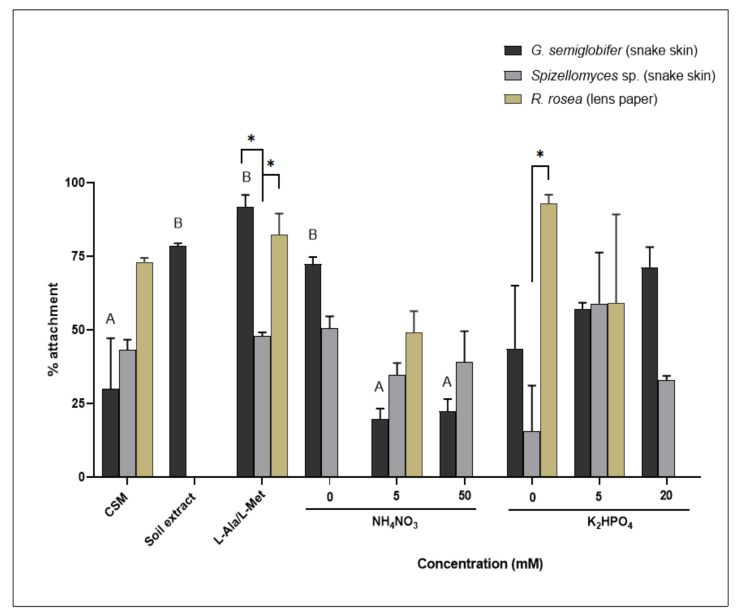
Percentage attachment of chytrid sporangia to baits (snake skin or lens paper). Error bars are SEM. Significant differences were observed between the nitrogen treatments (F_5, 36_ = 32.19, *p* < 0.05) and denoted by capital letters (A, B), but no significant changes were found in the phosphate series (F_4, 30_ = 3.343, *p* > 0.05). Significance found between isolates (F_2, 36_ = 18.08, *p* < 0.05) is denoted by an asterix (∗).

**Table 1 jof-08-00341-t001:** Chytrid strains selected for experimental investigation. Isolates were previously collected from sites in New South Wales. Initial identification was performed using morphological characterisation and later recorded in GenBank with partial 18s rRNA sequences [7,8]. Stock cultures are maintained on solid and liquid complex peptone, yeast and glucose (PYG) medium (2.5 mM peptone, 4.56 mM yeast extract, 27.7 mM glucose, ± 2% *w*/*v* agar).

Isolate	Order	Accession Number	Collection Site	Bait Substrate	Collector
*Gaertneriomyces semiglobifer* Mar C/C2	Spizellomycetales	FJ827645 FJ827701 FJ827738	Narrabri— cotton crop (−30°16′47.4″, 149°47′43.2″)	Pine pollen	Commandeur, Z.
*Spizellomyces* sp. Dec CC 4-10Z	Spizellomycetales	AB586075 AB586080	Narrabri— cotton fallow crop (−30°16′47.4″, 149°47′43.2″)	Pine pollen	Commandeur, Z.
*Rhizophlyctis rosea* AUS 13	Rhizophlyctidales	EU379156 EU379199	University of Sydney— garden soil (−33°53′23.8″, 151°11′31.4″)	Filter paper (cellulose)	Letcher, P.

**Table 2 jof-08-00341-t002:** Treatment media preparation. CSM is the positive control for all experiments. The treatments 0 mM nitrogen and 0 mM K_2_HPO_4_ are the negative controls for each of the nutrient concentration series.

	CSM	Starvation CSM	Organic Nitrogen (L-Ala/L-Met)	0 mM Nitrogen	NH_4_Cl Concentration Series	NaNO_3_ Concentration Series	0 mM K_2_HPO_4_	K_2_HPO_4_ Concentration Series
**MgSO_4_·7H_2_O (mM)**	1.66	1.66	1.66	1.66	1.66	1.66	1.66	1.66
**CaCl_2_ (mM)**	0.033	0.033	0.033	0.033	0.033	0.033	0.033	0.033
**NH_4_NO_3_ (mM)**	2.5	0	0	0	0	0	2.5	2.5
**NH_4_Cl (mM)**	0	0	0	0	1.5–50 ^1^	0	0	0
**NaNO_3_ (mM)**	0	0	0	0	0	1.5–50 ^1^	0	0
**D-glucose (mM)**	22	22	22	22	22	22	22	22
**L-alanine (mM)**	10.10	0	10.10	0	0	0	10.10	10.10
**L-methionine (mM)**	0.67	0	0.67	0	0	0	0.67	0.67
**K_2_HPO_4_ (mM)**	3.44	0	3.44	3.44	3.44	3.44	0	1.5–20 ^2^
**FeEDTA (μM)**	1.36	1.36	1.36	1.36	1.36	1.36	1.36	1.36
**Trace elements (mL) ^3^**	2.5	2.5	2.5	2.5	2.5	2.5	2.5	2.5
**DI H_2_O ^4^ (mL)**	997.5	997.5	997.5	997.5	997.5	997.5	997.5	997.5

^1^ 1.5, 3, 5, 10, 20, 50 mM series used for liquid media; ^2^ 5, 20, 50 mM concentration series used for solid media where agar (2% *w*/*v*) is added; ^3^ Trace element stock contains 10 μM MnCl_2_·4H_2_0, 10 μM ZnCl_2_, 33 μM H_3_BO_3_, 1 μM CuSO_4_.5H_2_O and 0.2 μM Na_2_MoO_4_·2H_2_O; ^4^ Deionised water (DI H_2_O).

## Data Availability

The data presented in this study are available on request from the corresponding authors. The authors acknowledge the technical and scientific assistance of Sydney Microscopy & Microanalysis, the University of Sydney node of Microscopy Australia.

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
