# Peer review of "The Effects of Nitrogen and Phosphorus on Colony Growth and Zoospore Characteristics of Soil Chytridiomycota"

_jof, 2022, doi:10.3390/jof8040341_

Round 1

Reviewer 1 Report

Some additional corrections are slightly made on the manuscript via PDF file. Furthermore, I have enclosed the comments which I hope will be helpful in revising your paper (i.e., a few of typos-linguistic corrections and suggested sentence as well as etc.) on the PDF file.

Author Response

Response to Reviewer 1 Comments

  1. Please delete these words. This is too much in detail. Your title should be “The effects of nitrogen and phosphorus on colony growth and zoospore characteristics of soil Chytridiomycota”

Change has been made.

  1. Suggested sentence: >>understanding of three soil chytrid species

Suggested change has been made

  1. suggested: >> and provide new techniques

Suggested change has been made.

  1. Suggested: >> Chytrids have been reported to respond to a variety of nitrogen or phosphorus sources [3,4], including at the genus level, however the effects of increased nutrient supplies on different stages of the chytrid life cycle and growth have not been studied.

Suggested change has been made.

  1. Suggested sentence: >>> Low microbial diversity, in addition to nutrient availability, can be an essential indicator of poor soil health.

Suggested change has been made.

  1. Suggested: >> We hypothesized that when nutrient concentrations increase, chytrid biomass and zoospore production would increase, but that future increases in nutritional concentrations may have a negative influence on growth.

Suggested change has been made but the word ‘further’ was retained because it relates to the potential increase in nutrient concentration.

  1. Please provide the full name of the group on first mention in the text, followed by the abbreviation. NSW >>>> New South Wales (NSW)

Change has been made.

  1. Please provide the GPS coordinates of an address or a place, simply use exact latitude and longitude position.

GPS coordinates added.

  1. Suggested: >> was performed using

Change has been made.

  1. Which partial sequence? ITS data?

18s rRNA – this information has been included in the text.

  1. >> “ )”

Change has been made.

  1. numbers >> number

Change has been made.

  1. Please provide the GPS coordinates of an address or a place, simply use exact latitude and longitude position.

GPS coordinates added.

  1. Spell out a number—or the word number—when it occurs at the beginning of a sentence.

Change has been made – the number (weight) was placed in parentheses (100 g).

  1. 1 mL >>> One mL

Change has been made – the number (volume) was placed in parentheses (1 mL).

  1. rosea >>> Rhizophlyctis rosea

Change has been made.

  1. Please change: R. rosea >>> Rhizophlyctis rosea

Change has been made.

  1. rosea >>> Rhizophlyctis rosea

Change has been made.

  1. semiglobifer >>> Gaertneriomyces semiglobifer

Change has been made.

  1. Suggested: >>>No significant changes in lipid content (mean fluorescence) and motility (velocity, displacement, and meandering index) were observed when G. semiglobifer and R. rosea zoospores were cultured in different nitrogen treatment media.

Change has been made.

  1. Suggested: >> Alternative chytrid experimental methods are also presented, including the use of the fluorescent dye Nile red to stain chytrid lipids and track zoospore motility (Figure 4) and a technique for measuring chytrid biomass by total protein quantification.

Change has been made.

  1. Suggested: >>> ergosterol quantification is used to determine the biomass of fungal colony 370, however this sterol is absent from chytrid cell membranes.

Change has been made.

  1. Suggested: These findings of soil chytrids in the current study highlight not only differences in adaptations among the three species, but also shifts in each chytrid's strategic responses.

Change has been made.

  1. To support the points, please kindly provide reference(s) (citation(s)).

Reference included.

  1. Suggested: >>> Although zoospore lipid content cannot be correlated with spore size [57], the changes in lipid fluorescence and therefore lipid content observed in the current study in comparison to zoospore formation suggest a similar trade-off.

Change has been made.

  1. To support the points citation, please kindly provide reference(s) (citation(s))

Reference included.

Reviewer 2 Report

METHODS
Lines 110-111: "Three species of zoosporic true fungi isolated from soils in NSW were selected for this study."

Why these three species have been selected? Please, describe the criteria of selection.

Lines 235-236: "Statistical analyses were done using two-way analysis of variance (ANOVA) and Tukey’s correction."

Please, consider advanced statistical methods. I suspect, that ANOVA only may not be sufficient.

Lines 433-434: "It is likely that chytrid populations are limited in uncultivated Australian soils where inorganic phosphate salts are scarce."

Please, compare your results with other studies of chytrids, e.g. from Europe/US, if available. 

Lines 436-437: "More work is required to understand the intricacies and diversity of nitrogen and phosphorus metabolism by chytrids."

I suppose the general recommendations for the scientific community would be advisable. Studies on soil microbial communities rarely involve the chytrids. 

Lines 460; 377; 75-77: "Populations of arbuscular mycorrhizal fungi (AMF) tend to be discouraged by mineral fertilisation and their associations with plants may even be antagonised."

I am not sure, that the comparison of arbuscular mycorrhizal fungal symbionts to chytrids is legitimate. Moreover, please consider carefully the sentence "AMF (...) associations with plants may even be antagonised".

Author Response

Response to Reviewer 2 Comments

  1. Lines 110-111: "Three species of zoosporic true fungi isolated from soils in NSW were selected for this study."

    Why these three species have been selected? Please, describe the criteria of selection.

Changes have been made to address the Reviewer’s comment.

Text edited (Lines 113-118): “Three species of zoosporic true fungi from the University of Sydney Zoosporic fungi culture collection were chosen for this study. These isolates all originate from soils in New South Wales (NSW) and were selected for this study because they display similar growth cycles and highlight both intra and inter-genus differences in responses to nutrient treatments.”

  1. Lines 235-236: "Statistical analyses were done using two-way analysis of variance (ANOVA) and Tukey’s correction."

    Please, consider advanced statistical methods. I suspect, that ANOVA only may not be sufficient.

The aim was to understand how an individual species would respond to the different treatments. Since it was expected that differences between each species would be present, ANOVA was considered as a standard method to identify statistically significant sources of variation, within each species, of the main variables of interest.

  1. Lines 433-434: "It is likely that chytrid populations are limited in uncultivated Australian soils where inorganic phosphate salts are scarce."

    Please, compare your results with other studies of chytrids, e.g. from Europe/US, if available. 

No references were found. Changes to the text were made to address this.

Text edited (Lines 442-446): “It is thought that chytrid populations are limited in uncultivated soils where inorganic phosphate salts are scarce [51]. Whilst there are currently no available studies to confirm this, it could be expected that as phosphate fertiliser applications increase, so too will some monocentric chytrid species.”

  1. Lines 436-437: "More work is required to understand the intricacies and diversity of nitrogen and phosphorus metabolism by chytrids."

    I suppose the general recommendations for the scientific community would be advisable. Studies on soil microbial communities rarely involve the chytrids. 

We agree with the Reviewer’s statement and have edited the text to address this.

Text edited (Lines 446-450): “Although the abundance of Chytridiomycota in soils is often low, more work is necessary if we want to understand the intricacies and diversity of chytrid nitrogen and phosphorus metabolism and hence determine their potential contributions to soil microbial ecology [30,31,33].”

  1. Lines 460; 377; 75-77: "Populations of arbuscular mycorrhizal fungi (AMF) tend to be discouraged by mineral fertilisation and their associations with plants may even be antagonised."

    I am not sure, that the comparison of arbuscular mycorrhizal fungal symbionts to chytrids is legitimate. Moreover, please consider carefully the sentence "AMF (...) associations with plants may even be antagonised".

Discussion of other fungi, including AMF, was used as examples to demonstrate certain points where available studies of chytrids are limited or could not be found. Alterations to the text have been made to highlight the use of AMF as an example for a discussion point, rather than a direct comparison to chytrids (Lines 74-78; 472-473).

Text edited (Lines 74-78): “Fertiliser additions can reduce the abundance and richness of fungal communities [27] with the potential to increase the abundance of plant pathogenic fungi [2], whilst populations and the diversity of other plant beneficial fungi, such as arbuscular mycorrhizal fungi (AMF), may instead decrease [28,29].”